# CERTIFIED $\ell_2$ ATTRIBUTION ROBUSTNESS VIA UNIFORMLY SMOOTHED ATTRIBUTIONS

## ABSTRACT

Model attribution is a popular tool to explain the rationales behind model predictions. However, recent work suggests that the attributions are vulnerable to minute perturbations, which can be added to input samples to manipulate the attributions while maintaining the prediction outputs. Although empirical studies have shown positive performance via adversarial training, an effective certified defense method is eminently needed to understand the robustness of attributions. In this work, we propose to use uniform smoothing technique that augments the vanilla attributions by noises uniformly sampled from a certain space. It is proved that, for all perturbations within the attack region, the cosine similarity between uniformly smoothed attribution of perturbed sample and the unperturbed sample is guaranteed to be lower bounded. We also derive alternative formulations of the certification that is equivalent to the original one and provides the maximum size of perturbation or the minimum smoothing radius such that the attribution can not be perturbed. We evaluate the proposed method on three datasets and show that the proposed method can effectively protect the attributions from attacks, regardless of the architecture of networks, training schemes and the size of the datasets.

## 1 INTRODUCTION

The developments and wider uses of deep learning models in various security-sensitive applications, such as autonomous driving, medical diagnosis, and legal judgments, have raised discussions of the trustworthiness of these models. The lack of explainability of deep learning models, especially the recent popular large language models (Brown et al., 2020), has been one of the main concerns. Regulators have started to require the explainability of the AI models in some applications (Goodman & Flaxman, 2017), and the explainability of AI models has been one of the main focuses of the research community (Doshi-Velez & Kim, 2017). Model attributions, as one of the important tools to explain the rationales behind the model predictions, have been used to understand the decision-making process of the models. For example, medical practitioners use the explanations generated by attribution methods to assist them in making important medical decisions (Antoniadi et al., 2021; Hrinivich et al., 2023; Du et al., 2022). Similarly, in autonomous vehicles, the explainability helps to deal with potential liability and responsibility gaps (Atakishiyev et al., 2021; Burton et al., 2020) and people naturally requires the confirmation of the safety critical decisions. However, since these users often lack expertise in machine learning and technical details of attributions, there is a risk that the attributions have been manipulated without noticing. Attackers may specifically target attributions to mislead investigators, propagate false narratives, or evade detection, potentially leading to serious consequences. Consequently, practitioners could lose trust in these methods, resulting in their refusal to use these methods. Thus, a trustworthy application requires not only the predictions made by the AI models, but also its explainability produced from attributions. However, the attributions have also been shown recently to be vulnerable to small perturbations (Ghorbani et al., 2019). Similar to adversarial attacks, attribution attacks generate perturbations that can be added to input samples. These perturbations distort the attributions while maintaining unchanged prediction outputs. This misleadingly gives a false sense of security to practitioners who blindly trust the attributions. An effective defense method is emergently needed to protect the attributions from the attacks.

Unlike adversarial defense, which has been extensively investigated to mitigate the harm of adversarial attacks that using both empirical (Madry et al., 2018; Athalye et al., 2018; Carlini & Wagner,

2017) and certified (Cohen et al., 2019; Wong & Kolter, 2018; Yang et al., 2020; Lecuyer et al., 2019) defense methods, attribution defense is neglected. Almost all attribution protection works focus on adversarially training the model by augmenting the training data with manipulated samples to improve the robustness of attributions (Boopathy et al., 2020; Chen et al., 2019; Ivankay et al., 2020; Sarkar et al., 2021; Wang & Kong, 2022). Levine et al. (2019) and a series of subsequent works (Liu et al., 2022; Huai et al., 2022; Gu et al., 2023) study the certifications of attributions under categorical ranking measurements, which are not easy to extend to other domains. A recent work by Wang & Kong (2023) attempts to derive a practical upper bound of cosine similarity for the worst-case attribution deviation, while suffering from strict assumptions and heavy computations. Overall, there is a gap in scalable methods for providing generalized certification of attribution robustness. In this work, we put our focus on the smoothed version of attributions and seek to provide a theoretical guarantee that the attributions are robust to any type of perturbations within $\ell_2$ attack budget.

Based on the previously defined formulation of attribution robustness (Wang & Kong, 2023), given a network $f$, its attribution function $g$ and perturbation $\boldsymbol{\delta} \in \mathbb{R}^d$, the attribution robustness is defined as the optimal value that maximizes the worst-case attribution difference $D(\cdot, \cdot)$ under provided attack budget,

$$\max_{\boldsymbol{\delta}} \quad D(g(\boldsymbol{x}), g(\boldsymbol{x} + \boldsymbol{\delta})) \quad \text{s.t.} \quad \|\boldsymbol{\delta}\| \le \epsilon. \tag{1}$$

However, the aforementioned study only provides an approximate method to solve the optimization problem under the strict assumptions that the networks is locally linear, and, as a result, is unable to generalize to modern neural networks. To give a complete certification on the problem, we follow the formulation and attempt to find the effective upper bound of the attribution difference, equivalently, the lower bound of attribution similarity, which can be applied to any network. We propose to use uniformly smoothed attribution, which is a smoothed version of the original attribution, and show that, for all perturbations within the allowable attack budget, the cosine similarity that measures the difference between perturbed and unperturbed uniformly smoothed attribution can be certified to be lower bounded. The contribution of this paper can be summarized as follows:

- We provide a theoretical guarantee that demonstrates the robustness of the uniformly smoothed attribution to any perturbations within allowable region. The robustness is measured by the similarity between the perturbed and unperturbed smoothed attribution. The method can be generally applied to any neural networks, and can be efficiently scaled to larger size images. To the best knowledge of the authors, this is the first work that provides a theoretical guarantee for attribution robustness.

- We present alternative formulations of the certification that are equivalent to the original one and also practical to be implemented. The alternative formulations determine the maximum size of perturbation, or the minimum radius of smoothing, ensuring that the attribution remains within a given tolerance.

- We demonstrate that the uniform smoothing can protect the attribution against $\ell_2$ attacks. More importantly, we evaluated the proposed method on the well-bounded integrated gradients and show that it can be effectively implemented and can successfully protect and certify the attributions from $\ell_2$ attacks.

The rest of this paper is organized as follows. In Section 2, we review the related works. In Section 3 and Section 4, we introduce the uniformly smoothed attribution and show that it can be certified against attribution attacks. In Section 5, we present experimental results and evaluate the proposed method. Finally, we conclude this paper and in Section 6.

## 2 RELATED WORKS

### 2.1 ATTRIBUTION METHODS

Attribution methods study the importance of each input feature, and measure that how much every feature contributes to the model prediction. One of the most popular attribution approaches is the gradient-based method. Based on the property that gradient is the measurement of the rate of change, the gradient-based methods measure the feature importance by weighting the gradients in different

ways. Examples of gradient-based methods include saliency (Simonyan et al., 2014), integrated gradients (IG) (Sundararajan et al., 2017), full-gradient (Srinivas & Fleuret, 2019), and *etc.* Other attribution methods include occlusion (Simonyan et al., 2014), which measures the importance of each input feature by occluding the feature and measuring the change of the model prediction, layer-wise relevance propagation (LRP) (Bach et al., 2015), which propagates the output relevance to the input layer, and SHAP related methods (Lundberg & Lee, 2017; Sundararajan & Najmi, 2020; Kwon & Zou, 2022). An important property of many attribution methods is the axiom of completeness that $\sum_i g_i(\boldsymbol{x}) = f_j(\boldsymbol{x})$, which indicates the relationship between attribution and the prediction score. Note that the gradient-based attribution methods are upper-bounded since the gradients of the model output with respect to the input are upper-bounded.

## 2.2 ATTRIBUTION ATTACKS AND DEFENSES

Ghorbani et al. (2019) first pointed out that attributions can be fragile to iterative attribution attack, and Dombrowski et al. (2019) extended the attack to be targeted that attributions can be changed purposely into any preset patterns. Similar to adversarial attacks, attribution attacks maximize the loss function that measures the difference between the original attributions and the target attributions. In addition, the attribution attacks are controlled not to alter the classification results. To defend against attribution attacks, adversarial training (Madry et al., 2018) approaches have been adopted. Chen et al. (2019) and Boopathy et al. (2020) minimizes the $\ell_p$-norm differences between perturbed and original attributions, and Ivankay et al. (2020) considers Pearson's correlation coefficient. It is worth noting that these methods empirically improve attribution robustness. Meanwhile, a series of certification methods Levine et al. (2019); Liu et al. (2022); Huai et al. (2022); Gu et al. (2023) have been proposed to ensure that attribution changes do not exceed a certain threshold under any perturbations within the allowable attack region. These methods measure attribution changes using *top-k intersection*, while a continuous alternative, cosine similarity, remains unexplored. It has been proved that using cosine similarity to measure attribution differences is consistent as top-k intersection and Kendall's rank correlation (Wang & Kong, 2022), and it is more likely to extend to other domains.

## 2.3 RANDOMIZED SMOOTHING

The smoothing technique has been popular in improving certified adversarial robustness (Liu et al., 2018; Lecuyer et al., 2019; Cohen et al., 2019; Yang et al., 2020). The smoothed classifiers take a batch of inputs that are randomly sampled from the neighbourhood of original inputs under certain distributions $\mu$ and make the decisions based on the most likely outputs, *i.e.*, $\arg\max_y \mathbb{P}_{\boldsymbol{\eta}\sim\mu}[F(\boldsymbol{x}+\boldsymbol{\eta})=y]$. They provide the certification of the a radius such that no perturbation within the radius can alter the classification label. Cohen et al. (2019) certifies the $\ell_2$ attack based on the Neyman-Pearson lemma and the result is alternatively proved by Salman et al. (2019) using explicit Lipschitz constants. Lecuyer et al. (2019) and Teng et al. (2020) consider Laplacian smoothing for the $\ell_1$ attack. Yang et al. (2020) derived a similar result in $\ell_\infty$ though the radius becomes small when the dimension of data gets large. Kumar & Goldstein (2021) applied randomized smoothing to structured output, which the attributions belong to, but the specific bound for attribution is too loose to be meaningful. Thus, there are no existing works that provide defense and valid certifications for attribution using randomized smoothing due to the difficulty of defining the attribution robustness and the computation of the attribution gradient. In this work, an effective method to formulate the smoothed attribution robustness as a simple optimization problem is proposed to provide certifications against attribution attacks.

## 3 UNIFORMLY SMOOTHED ATTRIBUTION

Consider a classifier $f : \mathbb{R}^d \to [0, 1]^c$ that maps the input $\boldsymbol{x} \in \mathbb{R}^d$ to the softmax output $y \in [0, 1]$, and its attribution function $g(\boldsymbol{x}) : \mathbb{R}^d \to \mathbb{R}^d$. The *smoothed attribution* of $f$ is to construct a new attribution $h$ by taking the mean of attributions on $\boldsymbol{x} + \boldsymbol{\eta}$, where $\boldsymbol{\eta}$ is randomly drawn from a density $\mu$, *i.e.*, the smoothed attribution $h$ can be defined as follows:

$$h(\boldsymbol{x}) = \mathbb{E}_{\boldsymbol{\eta}\sim\mu}[g(\boldsymbol{x}+\boldsymbol{\eta})]. \tag{2}$$

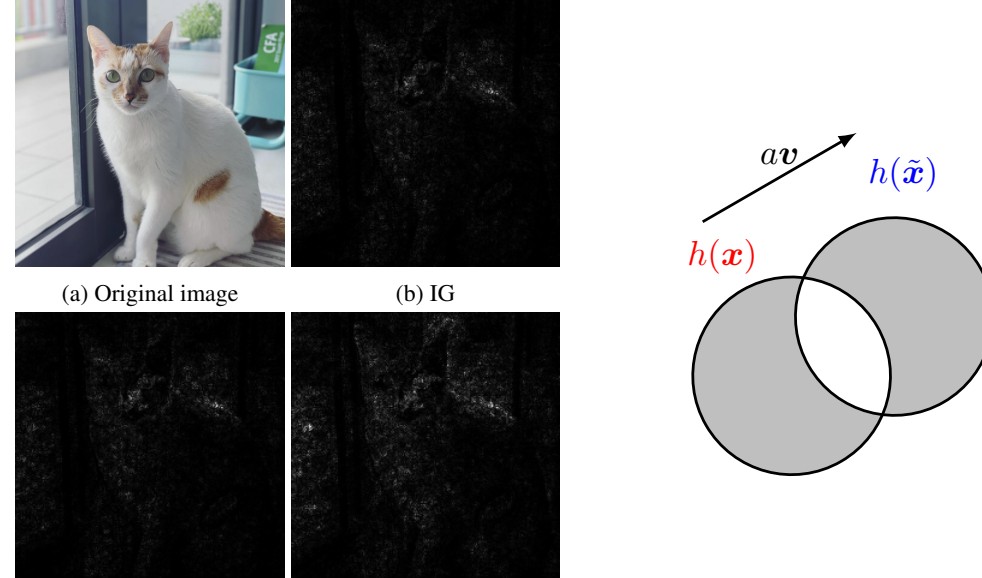

(a) Original image          (b) IG

(c) Gaussian smoothed IG    (d) Uniformly smoothed IG

Figure 1: (**left**) Examples of attributions (zoom in for better visibility). We choose to show the integrated gradients (IG) and its corresponding smoothing results. For Gaussian smoothing, the noise level is set to $\sigma = 0.2$ and for uniformly smoothed IG, $\ell_2$ ball with radius $\sqrt{3}\sigma$ is used. (**right**) A 2D illustration of the volumes of $\mathcal{B}(\boldsymbol{x}; r)$ and $\mathcal{B}(\tilde{\boldsymbol{x}}; r)$, as well as the relationship between $h(\boldsymbol{x})$ and $h(\tilde{\boldsymbol{x}})$. Here $h(\boldsymbol{x})$ is the original attribution, and $a\boldsymbol{v}$ represents the magnitude and direction of the translation of $h(\boldsymbol{x})$ after the sample is perturbed. $V_U$ in Theorem 1 is the volume of shaded region in the figure, and $V_S$ is the volume of each individual ball. When $h(\boldsymbol{x})$ is fixed, the lower bound of the cosine similarity between $h(\boldsymbol{x})$ and $h(\boldsymbol{x}) + a\boldsymbol{v}$ can be derived as a function of volumes

To construct smoothed attribution, the density $\mu$ can be chosen arbitrarily, and the smoothed attributions provide visually sharpened gradient-based attributions (see Figure 1 (left)). Smilkov et al. (2017) choose $\mu$ to be multivariate Gaussian distribution on input gradients to weaken the visually noisy attribution. In this work, we aim at certifying the attribution robustness under $\ell_2$ attack; thus we choose $\mu$ to be a uniform distribution on a $d$-dimensional closed space $\mathcal{S}$ centered at $\boldsymbol{0}$, especially the $\ell_2$-norm ball of radius $r$, $\mathcal{B}(\boldsymbol{0}; r) = \{\boldsymbol{y} : \|\boldsymbol{y}\|_2 \leq r, \boldsymbol{y} \in \mathbb{R}^d\}$. It can be seen that smoothing under this setting also provides high attribution quality (Figure 1d). To quantitatively evaluate the effectiveness of uniformly smoothed attribution, we can further evaluate its performance using GridPG introduced by Rao et al. (2022), which quantifies the significance of individual features in terms of positive contributions or influences. The GridPG values of IG, uniformly smoothed IG and Gaussian smoothed IG for $5,000$ randomly selected ImageNet examples are $0.4021$, $0.4093$ and $0.4110$, respectively, which suggest that the uniformly smoothed attributions can achieve comparable performance of GridPG with the Gaussian smoothed attributions, as well as the original non-smoothed attributions.

## 4    CERTIFYING THE COSINE SIMILARITY OF SMOOTHED ATTRIBUTIONS

We now consider $\mathcal{S}$ as an $\ell_2$-norm ball for the ease of analyzing the certification. Adapted from the formulation in Eq. (1), the robustness of attribution is defined as the minimum possible attribution similarity when a natural image is perturbed by attribution attacks. As mentioned in Section 2, this work studies cosine similarity as the measurement of similarity, as it has been shown to be the most suitable alternative to the non-differentiable Kendall's rank correlation, the most common evaluation index (Wang & Kong, 2022). Suppose that the $\ell_2$ attribution attack is performed upon input sample $\boldsymbol{x}$, the maximum allowable perturbation is $\epsilon$, *i.e.*, $\tilde{\boldsymbol{x}} = \boldsymbol{x} + \boldsymbol{\delta}$, where $\|\boldsymbol{\delta}\|_2 \leq \epsilon$. For all $\|\boldsymbol{\delta}\|_2 \leq \epsilon$, we want to find out the minimum value of $\cos(h(\boldsymbol{x}), h(\tilde{\boldsymbol{x}}))$, *i.e.*, the lower bound of cosine similarity

for attributions. Thus, the optimization problem we are interested in is formulated as follows:

$$\min_{\boldsymbol{\delta}} \quad \cos(h(\boldsymbol{x}), h(\boldsymbol{x} + \boldsymbol{\delta})) \quad \text{s.t.} \quad \|\boldsymbol{\delta}\|_2 \le \epsilon. \tag{3}$$

That is, we will show that, given an arbitrary sample point $\boldsymbol{x}$, for all perturbed samples $\tilde{\boldsymbol{x}}$, the cosine similarity between the original smoothed attribution and the corresponding perturbed smoothed attributions is guaranteed to be lower bounded by the optimum of (3). Alternatively, in a more practical perspective, given a threshold $T$ for the cosine similarity, we want to know the maximum size of perturbation, or the minimum smoothing radius, such that no perturbations inside would cause the attribution difference to exceed the threshold.

However, although optimizing the cosine similarity for attribution can be an intuitive way to find the lower bound, it is difficult in this problem to directly study the cosine function. Moreover, it is also intractable to optimize the cosine similarity with respect to a vector $\boldsymbol{\delta}$. To address this issue, we first reformulate the problem into an optimization over two scalars and then solve the alternative problem to obtain the lower bound of cosine similarity. All the proofs and derivations of theorems, lemmas and corollaries are provided in the Appendix A.

### 4.1 ONE-DIMENSIONAL REFORMULATION

We note that cosine similarity of attributions is in fact an inner product of their normalized vectors. Besides, we also observe that $h(\boldsymbol{x})$ is the mean of $g(\boldsymbol{x} + \boldsymbol{\eta})$ with respect to $\boldsymbol{\eta}$, which is, by definition, equivalent to the integral of $g$ weighted by the density of uniform distribution over $\mathcal{B}(\boldsymbol{x}; r)$. More importantly, for perturbed example $\tilde{\boldsymbol{x}}$, the weighting region is $\mathcal{B}(\tilde{\boldsymbol{x}}; r)$, which is a translated version of $\mathcal{B}(\boldsymbol{x}; r)$ and they are expected to intersect with each other when the distance of their centers is smaller than twice of the radius $r$. Therefore, given the input sample $\boldsymbol{x}$ and its attribution $h(\boldsymbol{x})$, we can rewrite the minimization of cosine similarity as follows:

$$\min_{a, \boldsymbol{v}} \quad \frac{h(\boldsymbol{x})^T}{\|h(\boldsymbol{x})\|} \left( \frac{h(\boldsymbol{x}) + a\boldsymbol{v}}{\|h(\boldsymbol{x}) + a\boldsymbol{v}\|} \right) \tag{4}$$

where $a$ represents the magnitude of the translation and the unit vector $\boldsymbol{v}$ is the direction of translation as shown in Figure 1 (right). It can be shown that the magnitude $a$ is constrained by a constant related to the intersecting volume and the property of the attribution function itself.

In a high-dimensional case, for a fixed cosine similarity value, a given $h(\boldsymbol{x})$ and $h(\tilde{\boldsymbol{x}})$, in fact, form a spherical cone. Thus, we can decompose the directional unit vector $\boldsymbol{v}$ into $\boldsymbol{v} = \cos\theta \boldsymbol{v}_{\|} + \sin\theta \boldsymbol{v}_{\perp}$, where $\boldsymbol{v}_{\perp}$ is perpendicular to $h(\boldsymbol{x})$ and $\boldsymbol{v}_{\|}$ is parallel to $h(\boldsymbol{x})$. Then, we have

$$\min_{a, \theta} \quad \frac{\|h(\boldsymbol{x})\| + a\cos\theta}{\sqrt{(\|h(\boldsymbol{x})\| + a\cos\theta)^2 + (a\sin\theta)^2}}, \tag{5}$$

where $0 \le \theta \le 2\pi$.

Since $a$ is a scalar representing the magnitude of the translation from $h(\boldsymbol{x})$ to $h(\tilde{\boldsymbol{x}})$, it can be shown that the magnitude of $a$ is upper bounded by the magnitude of the gradient weighted by the ratio of volume change during the translation process. Specifically, $a \le MV_U/V_S$, where $V_S$ is the volume of the $\ell_2$-ball $\mathcal{B}(\boldsymbol{0}; r)$, $V_U$ is the volume of the union of the two sampling space centered at $\boldsymbol{x}$ and $\tilde{\boldsymbol{x}}$ minus their intersection, and $M$ is a constant that depends on the upper bound of $g$, We notice that the gradient-based attribution is a function of input gradient, $\nabla f(\boldsymbol{x})$, which is bounded for Lipschitz continuous networks (See Lemma 1 in the Appendix A); thus, the upper bound can be derived separately for different attribution functions.

### 4.2 LOWER BOUND OF COSINE SIMILARITY FOR BOUNDED ATTRIBUTIONS

Now that we have reformulated the optimization with respect to vector $\boldsymbol{v}$ into an alternative simpler one with respect to scalar values $a$ and $\theta$. By solving the alternative problem, our result shows that the smoothed attribution is robust within the following half-angle of a spherical cone.

**Theorem 1.** *Let $g : \mathbb{R}^d \to \mathbb{R}^d$ be a upper bounded attribution function, and $\boldsymbol{\eta} \overset{U}{\sim} \mathcal{B}(\boldsymbol{0}; r)$. Let $h$ be the smoothed version of $g$ as defined in (2). Then, for all $\tilde{\boldsymbol{x}} \in \{\boldsymbol{x} + \boldsymbol{\delta} | \|\boldsymbol{\delta}\|_2 \le \epsilon\}$, we have*

Table 1: Comparison of top-k and Kendall's rank correlation between $\ell_2$ perturbed and non-perturbed attributions on standard and robust models using smoothed and non-smoothed attributions.

|          |         | Standard | IG-NORM (Chen et al., 2019) | TRADES (Zhang et al., 2019) | IGR (Wang & Kong, 2022) |
|----------|---------|----------|----------|----------|----------|
| SM       | top-k   | 0.3449   | 0.6575   | 0.6450   | 0.8354   |
|          | Kendall | 0.1496   | 0.4709   | 0.4642   | 0.7553   |
| SmoothSM | top-k   | 0.3853   | 0.6261   | 0.6082   | 0.8363   |
|          | Kendall | 0.1670   | 0.4238   | 0.4119   | 0.7568   |
| IG       | top-k   | 0.4742   | 0.7075   | 0.6821   | 0.8402   |
|          | Kendall | 0.1744   | 0.5098   | 0.5030   | 0.7839   |
| SmoothIG | top-k   | 0.5302   | 0.6730   | 0.6528   | 0.8460   |
|          | Kendall | 0.3819   | 0.4494   | 0.4533   | 0.7612   |

*$\cos(h(\boldsymbol{x}), h(\tilde{\boldsymbol{x}})) \geq T$, where*

$$T = \frac{\|h(\boldsymbol{x})\|_2}{\sqrt{\|h(\boldsymbol{x})\|_2^2 + M^2 V_U^2 / V_{\mathcal{S}}^2}} \tag{6}$$

*Here, $M$ is the upper bound of $g$. $V_{\mathcal{S}}$ is the volume of the $\ell_2$-ball $\mathcal{B}(\boldsymbol{0}; r)$, and $V_U$ is the volume of the union of the two sampling space centered at $\boldsymbol{x}$ and $\tilde{\boldsymbol{x}}$ minus their intersection.*

The entire proof can be found in Appendix A. The theorem points out that the lower bound of cosine similarity is related to the smoothing space around the input samples and the maximum allowable perturbation size. Moreover, when the smoothing space is a $\ell_2$-norm ball, the above result can be derived by directly computing two volumes $V_U$ and $V_{\mathcal{S}}$, which can be explicitly calculated by

$$V_U = 2V_{\mathcal{S}} \times \left(1 - I_{(2rh-h^2)/r^2}\left(\frac{d+1}{2}, \frac{1}{2}\right)\right) \tag{7}$$

where $h = r - \epsilon/2 \geq 0$ and $I_x(a, b)$ is the regularized incomplete beta function, cumulative density function of beta distribution (Li, 2010).

We observe the following properties of the above theorem.

1. Unlike previous attribution robustness works, such as Dombrowski et al. (2019), Singh et al. (2020), Boopathy et al. (2020) and Wang & Kong (2022), which require the networks to be twice-differentiable and need to change the ReLU activation into Softplus, the proposed result does not assume anything on the classifiers. Thus, it can be safely applied to any neural network and any architectures.

2. The lower bound depends on the radius of smoothing, $r$. The bound becomes larger as the radius of smoothing grows. In extreme cases when $r$ tends to infinity, the smoothing spaces of two samples completely overlap, which corresponds to the same smoothed attribution and their cosine similarity becomes $1$.

3. At the same time, the lower bound also decreases when the attack budget $\epsilon$ increases. Besides, when $\epsilon$ increases beyond the constraint that $h = r - \epsilon/2 \geq 0$ and tends to infinity, the distance between two attributions will become further and their cosine similarity will tend to $0$ in high dimensional space, which makes the lower bound becomes trivial.

4. The proposed method can be scaled to datasets with large images. The lower bound is efficient to compute since only the smoothed attribution of given sample needed. On the contrary, the previous works that approximately estimate the attribution robustness (Wang & Kong, 2023) require the computation of input Hessian and the corresponding eigenvalues and eigenvectors, which becomes intractable for larger size images on modern neural networks.

Table 2: The theoretical lower bound ($T$ in Eqn. (6)) for cosine similarity evaluated on baseline models using MNIST. Note that the bound is not achievable for $r = 0.5$ when $\epsilon = 1.0$, since the radius must be greater than $\epsilon/2$.

| $\epsilon = 0.5$ | $\ell_2$ radius ($r$) | 0.5 | 1.0 | 1.5 | 2.0 | 2.5 | 3.0 | 3.5 |
|---|---|---|---|---|---|---|---|---|
| | Standard | 0.3002 | 0.3141 | 0.3385 | 0.3732 | 0.4144 | 0.4600 | 0.5057 |
| | IG-NORM | 0.4038 | 0.4189 | 0.4432 | 0.4729 | 0.5055 | 0.5466 | 0.5909 |
| | IGR | 0.4145 | 0.4269 | 0.4482 | 0.4792 | 0.5208 | 0.5748 | 0.6392 |
| $\epsilon = 1.0$ | $\ell_2$ radius ($r$) | 0.5 | 1.0 | 1.5 | 2.0 | 2.5 | 3.0 | 3.5 |
| | Standard | / | 0.3034 | 0.3092 | 0.3264 | 0.3716 | 0.3990 | 0.4178 |
| | IG-NORM | / | 0.3650 | 0.3822 | 0.3892 | 0.4220 | 0.4974 | 0.5365 |
| | IGR | / | 0.3834 | 0.4025 | 0.4558 | 0.4914 | 0.5237 | 0.5358 |

### 4.3 Alternative formulations of the attribution robustness

The previous section formulates the robustness of smoothed attribution in terms of the smallest cosine similarity between the original and perturbed smoothed attribution, when the attack budget and the smoothing radius are fixed. In some scenarios, practitioners want to formulate the robustness in different ways. For example, we may want to find the maximum allowable perturbation $\epsilon$ such that the cosine similarity between the original and perturbed smoothed attribution is guaranteed to be greater than a predefined threshold $T$. On the other hand, one can also obtain the minimum smoothing radius needed such that the desired attribution robustness is achieved within allowable attack region. The following corollary provides the alternative formulations of the attribution robustness.

**Corollary 1.** *Let $g : \mathbb{R}^d \to \mathbb{R}^d$ be a bounded attribution function, and $\boldsymbol{\eta} \overset{U}{\sim} \mathcal{B}(\boldsymbol{x}; r)$. Let $h$ be the smoothed version of $g$ as defined in (2).*

*(i) Given a predefined threshold $T \in [0, 1]$, then for all $\|\boldsymbol{\delta}\|_2 \leq \epsilon$, we have $\cos(h(\boldsymbol{x}), h(\boldsymbol{x} + \boldsymbol{\delta})) \geq T$, where*

$$\epsilon = 2r \sqrt{1 - I_Z^{-1}\left(\frac{d+1}{2}, \frac{1}{2}\right)}. \tag{8}$$

*(ii) Given a predefined threshold $T \in [0, 1]$ and the maximum perturbation size $\epsilon \geq 0$, the smoothed attribution satisfies $\cos(h(\boldsymbol{x}), h(\tilde{\boldsymbol{x}})) \geq T$ for all $\tilde{\boldsymbol{x}} \in \{\boldsymbol{x} + \boldsymbol{\delta} | \|\boldsymbol{\delta}\|_2 \leq \epsilon\}$ when $r \geq R$, where*

$$R = \frac{\epsilon}{2}\left(1 - I_Z^{-1}\left(\frac{d+1}{2}, \frac{1}{2}\right)\right)^{-\frac{1}{2}}. \tag{9}$$

$I_z^{-1}(a, b)$ *is the inverse of the regularized incomplete beta function, and $Z$ is defined as*

$$Z = 1 - \frac{\|h(\boldsymbol{x})\|_2}{2M}\left(\frac{1}{T^2} - 1\right) \tag{10}$$

The derivation of the Corollary can be found in Appendix A. We notice that, in these formulations, when the smoothing radius is larger, the maximum allowable perturbation is also larger, which allows stronger attacks while keeping the attribution similarity within a controllable range. Similarly, when the maximum allowable perturbation is larger, the minimum smoothing radius is also larger, which means that the attribution similarity can be maintained with a larger smoothing radius.

## 5 Experiments and results

In this section, we evaluate the effectiveness of uniformly smoothed attribution. Following previous work on attribution robustness, we use the $\ell_2$ attribution attack adapted from the Iterative Feature Importance Attacks (IFIA) by Ghorbani et al. (2019). It is first shown that the uniformly smoothed

Table 3: Theoretical lower bounds evaluated on non-robust model and ImageNet using different radii $r$ and attack sizes $\epsilon$. Note that the method is only applicable when radius $r$ must be greater than $\epsilon/2$.

| $r$ | 0.5 | 1.0 | 1.5 | 2.0 | 2.5 | 3.0 | 3.5 |
|---|---|---|---|---|---|---|---|
| $\epsilon = 0.5$ | 0.2612 | 0.2594 | 0.2704 | 0.2716 | 0.2892 | 0.2973 | 0.3029 |
| $\epsilon = 1.0$ | / | 0.1803 | 0.1992 | 0.1746 | 0.1904 | 0.2127 | 0.2502 |
| $\epsilon = 2.0$ | / | / | 0.1753 | 0.1852 | 0.2044 | 0.2015 | 0.2045 |

attributions are less likely to be perturbed comparing with non-smoothed attributions when being attacked by IFIA. After which, we present the results of certification using the proposed lower bound of cosine similarity. The experiments are conducted on baseline models including adversarial robust models(Madry et al., 2018; Zhang et al., 2019), attributional robust models(Chen et al., 2019; Singh et al., 2020; Ivankay et al., 2020; Wang & Kong, 2022), as well as non-robust models trained for standard classification tasks. Following those baseline models, the method is tested on the validation sets of MNIST (LeCun et al., 2010) using a small-size convolutional network, on CIFAR-10 (Krizhevsky, 2009) using a ResNet-18 (He et al., 2016), and on ImageNet (Russakovsky et al., 2015) using a ResNet-50 (He et al., 2016). More details of the experiments are described in the Appendix. All experiments are run on NVIDIA GeForce RTX 3090.[1]

To empirically compute the uniformly smoothed attribution for every sample, $N$ points are randomly sampled from the $d$-dimensional sphere uniformly, and augmented to the input sample. To do so, the sampling technique introduced by Box & Muller (1958) is applied. The integration in Eqn. 2 is then empirically estimated using Monte Carlo integration where the computation scales linearly with the number of samples, i.e., $\hat{h}(\boldsymbol{x}) = \frac{1}{N} \sum g(\boldsymbol{x} + \boldsymbol{\eta}_i)$, for $\boldsymbol{\eta}_i \overset{U}{\sim} \mathcal{B}(\boldsymbol{0}; r)$. For large $N$, the estimator $\hat{h}(\boldsymbol{x})$ almost surely converges to $h(\boldsymbol{x})$ (Feller, 1991); hence the convergence of $\hat{T}$ to $T$ can be obtained (see Appendix C.1 for details). Unless specifically stated, we choose the number of samples $N$ to be $100,000$ to compute the proposed lower bound, and $N^* = 300$ for the uniformly smoothed attribution being attacked in all experiments.

## 5.1 EVALUATION OF THE ROBUSTNESS OF UNIFORMLY SMOOTHED ATTRIBUTION

We first conduct the experiment to verify that the uniformly smoothed attribution itself is more robust than the original attribution. The uniform smoothing around the $\ell_2$ ball with radius 0.5 is applied to the saliency map (SM) (Simonyan et al., 2014) and integrated gradients (IG) (Sundararajan et al., 2017) and evaluate on CIFAR-10. The resultant attributions are denoted by SmoothSM and SmoothIG, respectively. We then attack the attributions using the $\ell_2$ IFIA attack and evaluate the robustness using Kendall's rank correlation and top-k intersection (Ghorbani et al., 2019). The experiments are evaluated on both non-robust model (*Standard*) and robust models (*IG-NORM*, *TRADES*, *IGR*). Note that IFIA is directly performed on the smoothSM and smoothIG, instead of its original counterpart. Since the PGD-like attribution attack requires to take the derivative of the attribution to determine the direction of gradient descent, the double backpropagation is needed. Thus, it is necessary to replace the ReLU activation by the twice-differentiable Softplus during attack (Dombrowski et al., 2019). The results are shown in Table 1.

We observe that for the non-robust model, both SmoothSM and SmoothIG perform better than its non-smoothed counterparts in both metrics, which shows that the uniformly smoothed attribution itself is more resistant to the attribution attacks. For models that are specifically trained to defend against the attribution attacks using heuristic methods adapted from adversarial training, *e.g.*, IG-NORM, TRADES and IGR, the smoothed attributions show comparable robustness to the non-smoothed attribution. Moreover, we also notice that SmoothIG performs better than SmoothSM, especially for the non-robust models. This can be attributed to the fact that IG satisfies the axiom of completeness, which ensures that the sum of IG is upper-bounded by the model output. In addition, we also observed that the smoothing technique does not always enhance the robustness of attribution, as measured by top-k and Kendall's rank correlation. It is worth noting that Yeh et al. (2019) argued that randomized smoothing can reduce attribution sensitivities and consequently improve ro-

---

[1]Source code will be released later.

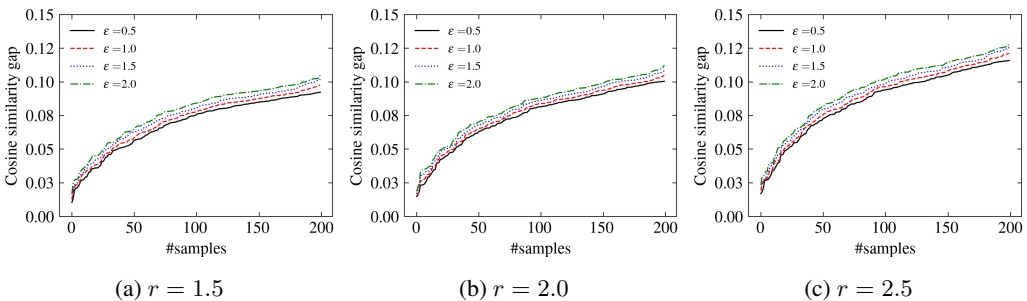

(a) $r = 1.5$ (b) $r = 2.0$ (c) $r = 2.5$

Figure 2: The gap between theoretical bounds and empirical cosine similarity between original and perturbed attribution evaluated on CIFAR-10 using IGR.

bustness. The disparity in our findings stems from the fact that we specifically evaluated Kendall's rank correlation, whereas their study defined sensitivity based on $\ell_2$-norm distance, which was subsequently deemed inappropriate for evaluating attribution robustness by Wang & Kong (2022).

### 5.2 EVALUATION OF THE CERTIFICATION OF UNIFORMLY SMOOTHED ATTRIBUTIONS

In this section, the lower bound of the cosine similarity between the original and attacked attribution is reported. We use the integrated gradients as an example since it is well-bounded due to the axiom of completeness, and the technique can be also applied to any other gradient-based attributions.

Table 2 reports the theoretical bound evaluated on MNIST computed using Theorem 1. We include the non-robust and two attributional robust models and compute the bound for different $\ell_2$ radius $r$ and attack size $\epsilon$ pairs. The lower bounds are validated by examining the actual $\ell_2$ attacks. Specifically, each input sample has been attacked 20 times and the cosine similarities of resulting perturbed attributions with original attributions are examined. In total, 200,000 attacked images are tested for each parameter pair and none of the evaluation metrics exceeds the theoretical bound. Moreover, we also observe that the bound becomes tighter when the $\ell_2$ radius $r$ increases and when the attack size $\epsilon$ decreases. Besides, since IGR is more robust than IG-NORM (Wang & Kong, 2022), we can observe that the lower bound is also a valid measurement of the robustness of the models.

In Figure 2, we show the gap between the theoretical lower bound and the empirical cosine similarity between the original and perturbed attributions. The results are evaluated on CIFAR-10 using IGR. Out of 10,000 testing samples, the 200 with the smallest gaps are chosen for each pair of $r$ and $\epsilon$, and the gaps are sorted for better visualization. We notice that the gaps between the theoretical bound and empirical cosine similarity are positive and small, which shows the validity and the tightness of the proposed bound.

In Table 3, we also include the theoretical lower bound evaluated on ImageNet to show that the proposed method is also applicable to large-scale datasets. Since the current attribution attacks and attribution defense methods do not scale to large-scale datasets, we only include the non-robust model. Since our method does not rely on the second-order derivative of the output with respect to the input, it can be scaled to ImageNet-size datasets. For the experiments on ResNet-50, each certification for one single sample takes around 15 seconds. We observe that the reported bounds are also consistent with our theoretical findings.

## 6 CONCLUSION

In this paper, we attempt to use the uniformly smoothed attribution to certify the attribution robustness evaluated by cosine similarity. The smoothed attribution is constructed by taking the mean of the attributions computed from input samples augmented by noises uniformly sampled from an $\ell_2$ ball. It is proved that the cosine similarity between the original and perturbed smoothed attribution is lower-bounded based on a geometric formulation related to the volume of the hyperspherical cap. Alternative formulations are provided to find the maximum allowable size of perturbations and the minimum radius of smoothing in order to maintain the attribution robustness. The method works

on bounded gradient-based attribution methods for all convolutional neural networks and is scalable to large datasets. We empirically demonstrate that the method can be used to certify the attribution robustness, using the well-bounded integrated gradients, and the state-of-the-art attributional robust models on MNIST, CIFAR-10 and ImageNet.

## 7 LIMITATIONS AND BROADER IMPACTS

The method in paper can be generally applied to any convolutional neural networks and any bounded attribution methods. Although the existence of an upper bound has been shown for all gradient-based methods, in some extreme cases when the upper bound for certain attribution is trivial, *i.e.*, an extremely large value, the proposed lower bound for attribution robustness also becomes trivial. In future work, we will investigate the upper bound for other bounded attribution methods and provide the corresponding lower bounds for attribution robustness. Besides, our current smoothing technique is restricted to the uniform distribution, and we will explore other distributions for the smoothing technique in future work.

Our work attempts to draw the attention of the community to the need for a guarantee of attribution robustness. With the increasingly large number of applications of deep learning, the transparency and trustworthiness of neural networks are crucial for users to understand the outcomes and to avoid any abuse of the techniques. While the study of the security of networks could reveal their potential risks that can be misused, we believe this work has more positive impacts to the community and can encourage the development of more trustworthy deep learning applications.

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

## A PROOFS

**Lemma 1.** *(Paulavičius & Žilinskas (2006)) For L-Lipschitz function $f : \mathbb{R}^d \to \mathbb{R}$,*

$$|f(\boldsymbol{x}) - f(\boldsymbol{y})| \leq L\|\boldsymbol{x} - \boldsymbol{y}\|_q \tag{11}$$

*where $L = \max\{\|\nabla f(\boldsymbol{x})\|_p : \boldsymbol{x} \in S\}$ is Lipschitz constant. Thus, $\|\nabla f(\boldsymbol{x})\|_p \leq L$.*

*Proof.* Refer to Paulavičius & Žilinskas (2006) for the proof. □

**Theorem 1.** *Let $g : \mathbb{R}^d \to \mathbb{R}^d$ be a upper bounded attribution function, and $\boldsymbol{\eta} \overset{U}{\sim} \mathcal{B}(\mathbf{0}; r)$. Let $h$ be the smoothed version of $g$ as defined in (2). Then, for all $\tilde{\boldsymbol{x}} \in \{\boldsymbol{x} + \boldsymbol{\delta} | \|\boldsymbol{\delta}\|_2 \leq \epsilon\}$, we have $\cos(h(\boldsymbol{x}), h(\tilde{\boldsymbol{x}})) \geq T$, where*

$$T = \frac{\|h(\boldsymbol{x})\|_2}{\sqrt{\|h(\boldsymbol{x})\|_2^2 + M^2 V_U^2/V_{\mathcal{S}}^2}} \tag{6}$$

*Here, $M$ is the upper bound of $g$. $V_{\mathcal{S}}$ is the volume of the $\ell_2$-ball $\mathcal{B}(\mathbf{0}; r)$, and $V_U$ is the volume of the union of the two sampling space centered at $\boldsymbol{x}$ and $\tilde{\boldsymbol{x}}$ minus their intersection.*

*Proof.* As defined in Eqn. (2)

$$h(\boldsymbol{x}) = \mathbb{E}_{\boldsymbol{\eta} \sim \mathcal{B}(\mathbf{0}; r)}[g(\boldsymbol{x} + \boldsymbol{\eta})] = \frac{1}{V_{\mathcal{S}}} \int_{\boldsymbol{\eta} \sim \mathcal{B}(\mathbf{0}; r)} g(\boldsymbol{x} + \boldsymbol{\eta}) d\boldsymbol{\eta} \tag{12}$$

where $V_{\mathcal{S}}$ is the volume of the $\ell_p$-ball with radius $r$. Similarly, let $\tilde{\boldsymbol{x}} = \boldsymbol{x} + \boldsymbol{\delta}$, where $\boldsymbol{\delta} \in \mathbb{R}^d$ is a vector and $\|\boldsymbol{\delta}\|_2 \leq \epsilon$. Then, we have

$$h(\tilde{\boldsymbol{x}}) = \frac{1}{V_{\mathcal{S}}} \int_{\boldsymbol{\eta} \sim \mathcal{B}(\mathbf{0}; r)} g(\tilde{\boldsymbol{x}} + \boldsymbol{\eta}) d\boldsymbol{\eta} \tag{13}$$

We note that when $\boldsymbol{\eta} \sim \mathcal{B}(\mathbf{0}; r)$, $\boldsymbol{x} + \boldsymbol{\eta} \sim \mathcal{B}(\boldsymbol{x}; r)$ and $\tilde{\boldsymbol{x}} + \boldsymbol{\eta} \sim \mathcal{B}(\tilde{\boldsymbol{x}}; r)$. We then rewrite $h(\boldsymbol{x})$ and $h(\tilde{\boldsymbol{x}})$ as follows:

$$h(\boldsymbol{x}) = \underbrace{\frac{1}{V_{\mathcal{S}}} \int_{\boldsymbol{x} \sim \mathcal{B}(\boldsymbol{x}; r) \setminus \mathcal{B}(\tilde{\boldsymbol{x}}; r)} g(\boldsymbol{x}) d\boldsymbol{x}}_{R_1} + \underbrace{\frac{1}{V_{\mathcal{S}}} \int_{\boldsymbol{x} \sim \mathcal{B}(\tilde{\boldsymbol{x}}; r) \cap \mathcal{B}(\boldsymbol{x}; r)} g(\boldsymbol{x}) d\boldsymbol{x}}_{R_2} \tag{14}$$

and

$$h(\tilde{\boldsymbol{x}}) = \underbrace{\frac{1}{V_{\mathcal{S}}} \int_{\boldsymbol{x} \sim \mathcal{B}(\tilde{\boldsymbol{x}}; r) \cap \mathcal{B}(\boldsymbol{x}; r)} g(\boldsymbol{x}) d\boldsymbol{x}}_{R_2} + \underbrace{\frac{1}{V_{\mathcal{S}}} \int_{\boldsymbol{x} \sim \mathcal{B}(\tilde{\boldsymbol{x}}; r) \setminus \mathcal{B}(\boldsymbol{x}; r)} g(\boldsymbol{x}) d\boldsymbol{x}}_{R_3} \tag{15}$$

Hence,

$$h(\tilde{\boldsymbol{x}}) = h(\boldsymbol{x}) - R_1 + R_3 \tag{16}$$

Denote $a\boldsymbol{v} = R_3 - R_1$, where $\boldsymbol{v}$ is a unit vector in the same direction of $R_3 - R_1$ and $a = \|R_3 - R_1\|_2$ is a scalar with the same magnitude of $R_3 - R_1$. Then, we have

$$\cos(h(\boldsymbol{x}), h(\tilde{\boldsymbol{x}})) = \frac{h(\boldsymbol{x})^\top}{\|h(\boldsymbol{x})\|_2} \left( \frac{h(\boldsymbol{x}) + a\boldsymbol{v}}{\|h(\boldsymbol{x}) + a\boldsymbol{v}\|_2} \right) \tag{17}$$

Note that the attribution $g(\boldsymbol{x})$ is upper bounded by $M$, specifically, $\|g(\boldsymbol{x})\|_2 \leq M$, for some constant $M$. Thus, we can derive that

$$a = \|R_3 - R_1\|_2 \tag{18}$$

$$= \left\| \frac{1}{V_{\mathcal{S}}} \left( \int_{\boldsymbol{x} \sim \mathcal{B}(\tilde{\boldsymbol{x}};r) \setminus \mathcal{B}(\boldsymbol{x};r)} g(\boldsymbol{x}) d\boldsymbol{x} - \int_{\boldsymbol{x} \sim \mathcal{B}(\boldsymbol{x};r) \setminus \mathcal{B}(\tilde{\boldsymbol{x}};r)} g(\boldsymbol{x}) d\boldsymbol{x} \right) \right\|_2 \tag{19}$$

$$\leq \frac{1}{V_{\mathcal{S}}} \left( \left\| \int_{\boldsymbol{x} \sim \mathcal{B}(\tilde{\boldsymbol{x}};r) \setminus \mathcal{B}(\boldsymbol{x};r)} g(\boldsymbol{x}) d\boldsymbol{x} \right\|_2 + \left\| \int_{\boldsymbol{x} \sim \mathcal{B}(\boldsymbol{x};r) \setminus \mathcal{B}(\tilde{\boldsymbol{x}};r)} g(\boldsymbol{x}) d\boldsymbol{x} \right\|_2 \right) \tag{20}$$

$$\leq \frac{1}{V_{\mathcal{S}}} \left( \int_{\boldsymbol{x} \sim \mathcal{B}(\tilde{\boldsymbol{x}};r) \setminus \mathcal{B}(\boldsymbol{x};r)} \|g(\boldsymbol{x})\|_2 \, d\boldsymbol{x} + \int_{\boldsymbol{x} \sim \mathcal{B}(\boldsymbol{x};r) \setminus \mathcal{B}(\tilde{\boldsymbol{x}};r)} \|g(\boldsymbol{x})\|_2 \, d\boldsymbol{x} \right) \tag{21}$$

$$\leq \frac{1}{V_{\mathcal{S}}} \left( \int_{\boldsymbol{x} \sim \mathcal{B}(\tilde{\boldsymbol{x}};r) \setminus \mathcal{B}(\boldsymbol{x};r)} M d\boldsymbol{x} + \int_{\boldsymbol{x} \sim \mathcal{B}(\boldsymbol{x};r) \setminus \mathcal{B}(\tilde{\boldsymbol{x}};r)} M d\boldsymbol{x} \right) \tag{22}$$

$$= M \times \frac{V_{\mathcal{B}(\boldsymbol{x};r) \setminus \mathcal{B}(\tilde{\boldsymbol{x}};r) \cup \mathcal{B}(\tilde{\boldsymbol{x}};r) \setminus \mathcal{B}(\boldsymbol{x};r)}}{V_{\mathcal{S}}} = M \frac{V_U}{V_{\mathcal{S}}} \tag{23}$$

Thus, the lower bound of $\cos(h(\boldsymbol{x}), h(\tilde{\boldsymbol{x}}))$ can be found by solving the optimization problem [2]

$$\min_{\boldsymbol{v}} \quad \frac{h(\boldsymbol{x})^\top}{\|h(\boldsymbol{x})\|} \left( \frac{h(\boldsymbol{x}) + a\boldsymbol{v}}{\|h(\boldsymbol{x}) + a\boldsymbol{v}\|} \right)$$

$$\text{s.t.} \quad \|\boldsymbol{v}\| = 1 \tag{24}$$

$$a \leq M \frac{V_U}{V_{\mathcal{S}}}$$

Since $h(\boldsymbol{x})$ and $h(\tilde{\boldsymbol{x}})$ form a spherical cone, we can decompose $\boldsymbol{v}$ by $\boldsymbol{v} = \cos\theta \boldsymbol{v}_\| + \sin\theta \boldsymbol{v}_\perp$, where $\boldsymbol{v}_\|$ and $\boldsymbol{v}_\perp$ are two orthogonal unit vectors such that $h^\top(\boldsymbol{x})\boldsymbol{v}_\perp = 0$ and $\boldsymbol{v}_\| = h(\boldsymbol{x})/\|h(\boldsymbol{x})\|$. Then, the optimization problem can be rewritten as

$$\min \quad \boldsymbol{v}_\|^\top \left( \frac{h(\boldsymbol{x}) + a(\cos\theta \boldsymbol{v}_\| + \sin\theta \boldsymbol{v}_\perp)}{\|h(\boldsymbol{x}) + a(\cos\theta \boldsymbol{v}_\| + \sin\theta \boldsymbol{v}_\perp)\|} \right) \tag{25}$$

$$\Rightarrow \min \quad \boldsymbol{v}_\|^\top \left( \frac{\|h(\boldsymbol{x})\|\boldsymbol{v}_\| + a(\cos\theta \boldsymbol{v}_\| + a\sin\theta \boldsymbol{v}_\perp)}{\|\|h(\boldsymbol{x})\|\boldsymbol{v}_\| + a(\cos\theta \boldsymbol{v}_\| + a\sin\theta \boldsymbol{v}_\perp)\|} \right) \tag{26}$$

$$\Rightarrow \min \quad \frac{(\|h(\boldsymbol{x})\| + a\cos\theta)\boldsymbol{v}_\|^\top \boldsymbol{v}_\| + a\sin\theta \boldsymbol{v}_\|^\top \boldsymbol{v}_\perp}{\sqrt{(\|h(\boldsymbol{x})\| + a\cos\theta)^2 \boldsymbol{v}_\|^\top \boldsymbol{v}_\| + (a\sin\theta)^2 \boldsymbol{v}_\perp^\top \boldsymbol{v}_\perp}} \tag{27}$$

$$\Rightarrow \min \quad \frac{\|h(\boldsymbol{x})\| + a\cos\theta}{\sqrt{(\|h(\boldsymbol{x})\| + a\cos\theta)^2 + (a\sin\theta)^2}} \tag{28}$$

Since $h(\boldsymbol{x})$ is known for a given sample, the optimization problem can be written as follows by taking $\|h(\boldsymbol{x})\| = c$:

$$\min \quad \frac{c + a\cos\theta}{\sqrt{(c + a\cos\theta)^2 + (a\sin\theta)^2}}$$

$$\text{s.t.} \quad a \leq M \frac{V_U}{V_{\mathcal{S}}} \tag{29}$$

We now consider the Lagrange function of the optimization problem:

$$\mathcal{L}(x, \theta, \lambda) = \frac{c + a\cos\theta}{\sqrt{(c + a\cos\theta)^2 + (a\sin\theta)^2}} - \lambda\left(a - M\frac{V_U}{V_{\mathcal{S}}}\right) \tag{30}$$

Taking the derivative of $\mathcal{L}$ with respect to $a$ and $\theta$ and setting them to zero, we have

$$\frac{\partial}{\partial a}\mathcal{L} = \frac{1}{T^2}\left(T\cos\theta - \frac{1}{T}(c\cos\theta + 2a) \times (c + a\cos\theta)\right) - \lambda = 0 \tag{31}$$

---

[2]$\|\cdot\|$ in the following content denotes the $\ell_2$-norm unless otherwise specified.

and

$$\frac{\partial}{\partial \theta}\mathcal{L} = \frac{1}{T^2}\left(-a\sin\theta \cdot T + \frac{1}{T}\left(c^2 a\sin\theta + ca^2\sin\theta\cos\theta\right)\right) = 0 \tag{32}$$

where $T = \sqrt{(c + a\cos\theta)^2 + (a\sin\theta)^2}$. Solving the above equations, we have

$$\cos\theta = 0 \quad \text{or} \quad a = 0 \tag{33}$$

where $a = 0$ reaches the maximum and $\cos\theta = 0$ is the minimum. Therefore, the lower bound of $\cos(h(\boldsymbol{x}), h(\tilde{\boldsymbol{x}}))$ is

$$\cos(h(\boldsymbol{x}), h(\tilde{\boldsymbol{x}})) \geq \frac{c}{\sqrt{c^2 + (M\frac{V_U}{V_S})^2}} = \frac{\|h(\boldsymbol{x})\|}{\sqrt{\|h(\boldsymbol{x})\|^2 + (MV_U/V_S)^2}} \tag{34}$$

$\square$

**Corollary 1.** *Let $g : \mathbb{R}^d \to \mathbb{R}^d$ be a bounded attribution function, and $\boldsymbol{\eta} \overset{U}{\sim} \mathcal{B}(\boldsymbol{x}; r)$. Let $h$ be the smoothed version of $g$ as defined in (2).*

*(i) Given a predefined threshold $T \in [0, 1]$, then for all $\|\boldsymbol{\delta}\|_2 \leq \epsilon$, we have $\cos(h(\boldsymbol{x}), h(\boldsymbol{x}+\boldsymbol{\delta})) \geq T$, where*

$$\epsilon = 2r\sqrt{1 - I_Z^{-1}\left(\frac{d+1}{2}, \frac{1}{2}\right)}. \tag{8}$$

*(ii) Given a predefined threshold $T \in [0, 1]$ and the maximum perturbation size $\epsilon \geq 0$, the smoothed attribution satisfies $\cos(h(\boldsymbol{x}), h(\tilde{\boldsymbol{x}})) \geq T$ for all $\tilde{\boldsymbol{x}} \in \{\boldsymbol{x} + \boldsymbol{\delta} \| \|\boldsymbol{\delta}\|_2 \leq \epsilon\}$ when $r \geq R$, where*

$$R = \frac{\epsilon}{2}\left(1 - I_Z^{-1}\left(\frac{d+1}{2}, \frac{1}{2}\right)\right)^{-\frac{1}{2}}. \tag{9}$$

$I_z^{-1}(a, b)$ *is the inverse of the regularized incomplete beta function, and $Z$ is defined as*

$$Z = 1 - \frac{\|h(\boldsymbol{x})\|_2}{2M}\left(\frac{1}{T^2} - 1\right) \tag{10}$$

*Proof.* Corollary 1 can be obtained by fixing $T$ and taking $r$ as unknown, and fixing $T$ and taking $\epsilon$ as unknown, respectively. We can first derive that

$$I_{(2rh-h^2)/r^2}\left(\frac{d+1}{2}, \frac{1}{2}\right) = 1 - \frac{\|h(x)\|_2}{2M}\sqrt{\frac{1}{T^2} - 1} = Z \tag{35}$$

Using the inverse of the regularized incomplete beta function, *i.e.*, $x = I_y^{-1}(a, b)$, and $h = r - \epsilon/2$, we have

$$I_Z^{-1}\left(\frac{d+1}{2}, \frac{1}{2}\right) = (2rh - h^2)/r^2 = 1 - \frac{\epsilon^2}{4r^2} \tag{36}$$

The results in Corollary can then be solved accordingly. $\square$

## B  IMPLEMENTATION DETAILS

In the experiments, we implemented the $\ell_2$ attribution attack adapted from Ghorbani et al. (2019). The attack uses top-$k$ intersection version as the loss function. Following previous works, we choose $k = 100$ for MNIST and $k = 1000$ for CIFAR-10. The number of iterations in PGD-like attack is 200, and the step size is 0.1. As mentioned in the main content, we do not implement the attack on ImageNet since the attribution attacks are not scalable to large size images. In the following parts of this section, we provide more details of evaluations in the experiments.

## B.1 ATTRIBUTION METHODS

We used saliency maps (SM) and integrated gradients (IG) in the evaluation sections. These two methods are defined as follows:

- Saliency maps: $\text{SM}(\boldsymbol{x}) = \frac{\partial f(\boldsymbol{x})}{\partial \boldsymbol{x}}$.

- Integrated gradients: $\text{IG}(\boldsymbol{x}) = (\boldsymbol{x} - \boldsymbol{x}') \times \int_{\alpha=0}^{1} \frac{\partial f(\boldsymbol{x}' + \alpha(\boldsymbol{x} - \boldsymbol{x}'))}{\partial \boldsymbol{x}} d\alpha$.

The SmoothSM and SmoothIG are the smoothed versions of SM and IG, respectively.

## B.2 EVALUATION METRICS

Given original attribution $g(\boldsymbol{x})$ and perturbed attribution $g(\tilde{\boldsymbol{x}})$, we use top-k intersection, Kendall's rank correlation (Ghorbani et al., 2019) and cosine similarity (Wang & Kong, 2022) to evaluate their differences.

- Top-k intersection measures the proportion of $k$ largest features that overlap between $g(\boldsymbol{x})$ and $g(\tilde{\boldsymbol{x}})$.

- Kendall's rank correlation measures the proportion of pairs of features that have the same order in $g(\boldsymbol{x})$ and $g(\tilde{\boldsymbol{x}})$: $\frac{2}{d(d-1)} \sum_{i=1}^{d} \sum_{j=i+1}^{d} \mathbf{1}_{\{g(\boldsymbol{x})_i > g(\boldsymbol{x})_j\}} \mathbf{1}_{\{g(\tilde{\boldsymbol{x}})_i > g(\tilde{\boldsymbol{x}})_j\}}$.

- Cosine similarity measures the cosine of the angle between $g(\boldsymbol{x})$ and $g(\tilde{\boldsymbol{x}})$: $\frac{g(\boldsymbol{x})^{\top} g(\tilde{\boldsymbol{x}})}{\|g(\boldsymbol{x})\| \|g(\tilde{\boldsymbol{x}})\|}$.

## B.3 BASELINE METHODS

We compare with the following adversarial and attributional robust models:

**IG-NORM (Chen et al., 2019)**

$$\text{CE}(f(\boldsymbol{x}), y) + \lambda \max_{\tilde{\boldsymbol{x}} \in \mathcal{B}_{\varepsilon}(\boldsymbol{x})} \|\text{IG}(\boldsymbol{x}, \tilde{\boldsymbol{x}})\|_1 \tag{37}$$

**TRADES (Zhang et al., 2019)**

$$\text{CE}(f(\tilde{\boldsymbol{x}}), y) + \beta \text{KL}(f(\boldsymbol{x}) \| f(\tilde{\boldsymbol{x}})) \tag{38}$$

**IGR (Wang & Kong, 2022)**

$$\text{CE}(f(\tilde{\boldsymbol{x}}), y) + \beta \text{KL}(f(\boldsymbol{x}) \| f(\tilde{\boldsymbol{x}})) + \lambda \left(1 - \cos(\text{IG}(\boldsymbol{x}), \text{IG}(\tilde{\boldsymbol{x}}))\right) \tag{39}$$

Here CE denotes the cross-entropy loss and KL denotes the Kullback-Leibler divergence.

# C ADDITIONAL EXPERIMENTS

## C.1 TEST ON MONTE CARLO ESTIMATION

Note that the bound given by Theorem 1 is deterministic. In this section, we provide a probabilistic bound for the attribution robustness. Specifically, we want to find the value of $t$ such that $Pr(T \leq t) = 1 - \alpha$, where $T$ is defined in Eqn. (6) and $\alpha$ is the significance level. Recall that $T$ is defined as follows:

$$T = \frac{\|h(\boldsymbol{x})\|_2}{\sqrt{\|h(\boldsymbol{x})\|_2^2 + c}} \tag{40}$$

where $c = M^2 V_U^2 / V_S^2$. If we denote that $Q = \|h(\boldsymbol{x})\|_2$, then we have

$$Pr(T \leq t) = Pr\left(\frac{Q}{\sqrt{Q^2 + c}} \leq t\right) = Pr\left(Q^2 \leq \frac{ct^2}{1 - t^2}\right) \tag{41}$$

Table 4: Evaluation of center smoothing on attributions

| $\epsilon_1$ | 0.1 | 0.2 | 0.3 | 0.4 | 0.5 |
|---|---|---|---|---|---|
| SmoothSM | 1.207 | 1.729 | 1.843 | 1.907 | 1.998 |

Note that we used Monte Carlo Integration to calculate the integral in $h(\boldsymbol{x})$, which estimates $h(\boldsymbol{x})$ by sampling $\boldsymbol{\eta}$ from $\mathcal{B}$, *i.e.*,

$$\hat{h}(\boldsymbol{x}) = \frac{1}{N} \sum_{i=1}^{N} g(\boldsymbol{x} + \boldsymbol{\eta}_i), \quad \boldsymbol{\eta}_i \sim \mathcal{B}. \tag{42}$$

Note that $\hat{h}(\boldsymbol{x})$ is an unbiased estimator of $h(\boldsymbol{x})$, *i.e.* $\mathbb{E}[\hat{h}(\boldsymbol{x})] = h(x)$. The estimator almost surely converges to $h(\boldsymbol{x})$ as $N \to \infty$, *i.e.* $\lim_{N \to \infty} \hat{h}(\boldsymbol{x}) = h(\boldsymbol{x})$ almost surely. By the Central Limit Theorem, the estimator $\hat{h}(\boldsymbol{x})$ has the following asymptotic distribution,

$$\hat{h}(\boldsymbol{x}) \overset{a.s.}{\sim} \mathcal{N}(h(\boldsymbol{x}), D), \tag{43}$$

which the covariance matrix $D = \text{diag}(\sigma_{ii}^2/N)$ can be estimated by the empirical variances of $g(\boldsymbol{x} + \boldsymbol{\eta}_i)$. Thus, the quadratic form $Q^2 = \|\hat{h}(\boldsymbol{x})\|_2^2$ can be seen as generalized chi-square distributed. We can derive the cumulative distribution function of Monte Carlo estimator $T_{MC}$ at $t$ as the cumulative distribution function of the generalized chi-square distribution at $\frac{ct^2}{1-t^2}$, *i.e.*,

$$Pr(T_{MC} \leq t) = F\left(\frac{ct^2}{1-t^2}\right), \tag{44}$$

where $F$ is the cumulative distribution function of the generalized chi-square distribution constructed from the quadratic form of Gaussian random variable with mean $h(\boldsymbol{x})$ and covariance $D$ (Davies, 1980; Das & Geisler, 2021). In this work, we use the R package CompQuadForm (Duchesne & De Micheaux, 2010) to compute the cumulative distribution function. For any fixed image sample $\boldsymbol{x}$, we can validate $t_2 - t_1$ is close to 0 when $Pr(t_1 \leq T_{MC} \leq t_2) = 1 - \alpha$ by solving the following equation. For small $\alpha = 0.01$ and the number of samples $N = 100,000$, we found that the values of $t_2 - t_1$ are at scale of $10^{-4}$ in MNIST and CIFAR-10, and $10^{-3}$ in ImageNet calculated by choosing $10,000$ samples from each dataset. This validates the error from Monte Carlo integral is minute and that the probabilistic bound is close to the deterministic bound.

$$F\left(\frac{ct_2^2}{1-t_2^2}\right) = 1 - \alpha/2 \quad \text{and} \quad F\left(\frac{ct_1^2}{1-t_1^2}\right) = \alpha/2. \tag{45}$$

### C.2 ADDITIONAL VISUALIZATION OF THE UNIFORMLY SMOOTHED ATTRIBUTIONS

In Figure 1 (left), we have shown that the uniformly smoothed attributions have a comparable quality as the original attributions. Here more examples are provided in Figure 3 to illustrate the quality of the uniformly smoothed attributions.

### C.3 EVALUATION OF CENTER SMOOTHING (KUMAR & GOLDSTEIN, 2021) ON ATTRIBUTIONS

To compare the performance with center smoothing (Kumar & Goldstein, 2021), we also implemented the same method to evaluate the certification of attributions. Specifically, we compute the bound for SmoothSM on IG-NORM using MNIST, and follow the same setting by choosing $h = 1$ and $\epsilon_1 = 0.1, 0.2, \cdots, 0.5$. Directly using the cosine similarity on the method is not applicable since cosine similarity does not satisfy the triangle inequality. Following the relaxation method in Sec.4 of Kumar & Goldstein (2021), a multiplier $\gamma = 2$ is added. Besides, we use $1 - \cos\theta$ to reflect the distance metric instead of the similarity metric. The results are shown in the Table 4. It can be observed that the upper bound for $1 - \cos\theta$ is greater than 1 for all the choices of $\epsilon$, which is trivially valid for the trigonometric function since we only consider $\cos\theta \in [0, 1]$. Thus, the upper bound provided in the aforementioned work can be too loose on our setting.

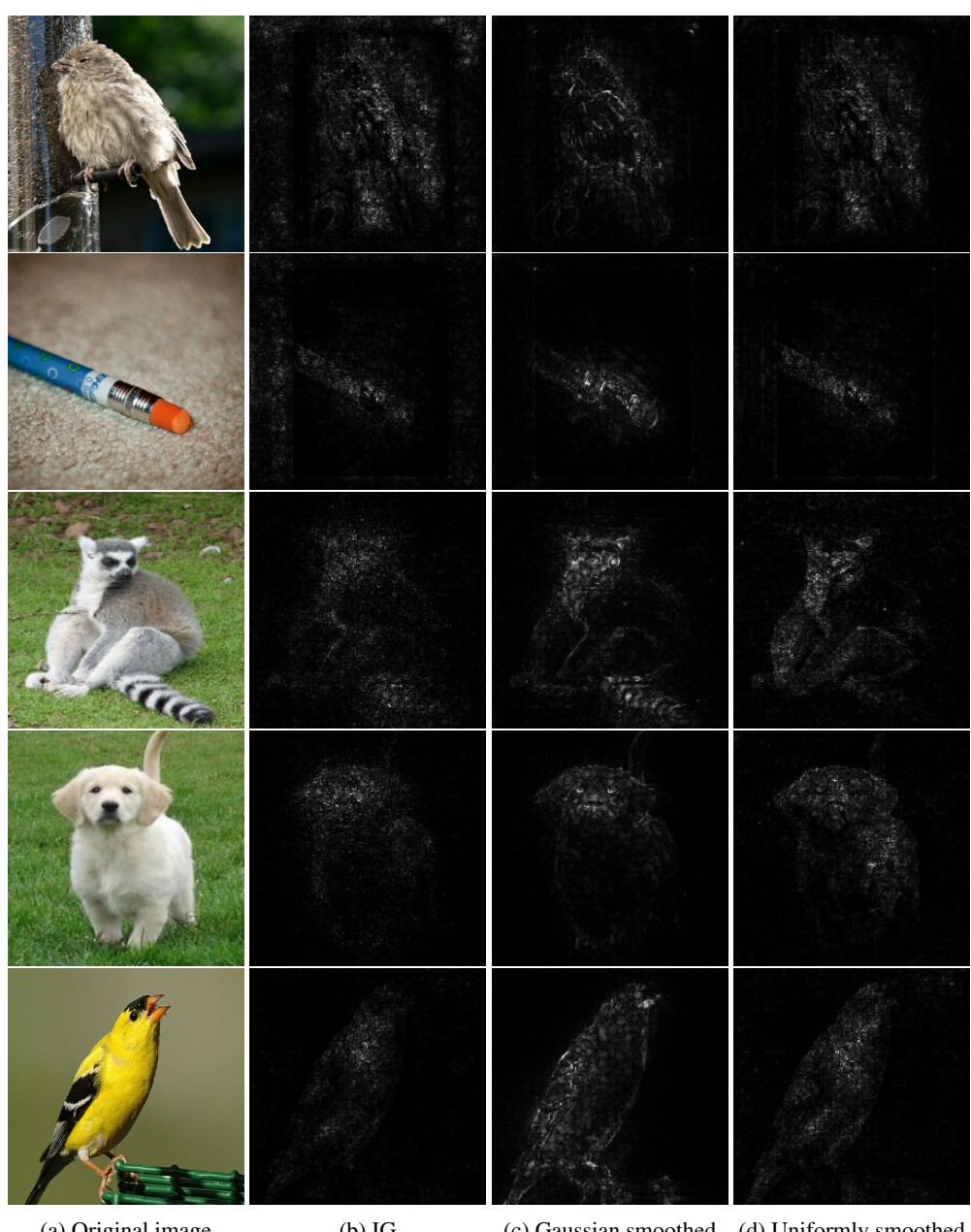

(a) Original image      (b) IG      (c) Gaussian smoothed      (d) Uniformly smoothed

Figure 3: Additional visualization of the attribution maps of the (a) original image, (b) IG, (c) Gaussian smoothed IG, and (d) uniformly smoothed IG.

## C.4 EVALUATION OF ALTERNATIVE FORMULATIONS

In Section 4.3, we introduced two alternative formulations of the proposed method that can be applied in specific scenarios. In this section, we provide additional information to report the experiments on these two formulations.

In Tables 5 to 7, which correspond to MNIST, CIFAR-10 and ImageNet, respectively, we report the computed values of the maximum allowable perturbation size. Under the size constraint, no examples can be found by the attacks against uniformly smoothed IG of a certain radius such that the cosine similarity between clean and perturbed attributions exceeds the given threshold ($T = 0.8$ and $T = 0.9$). The results are consistent with our theory. For larger radius smoothing, the maximum

Table 5: Maximum allowable perturbation size for different threshold ($T = 0.8$ and $T = 0.9$) under various choices of $\ell_2$ smoothing radii $r$ evaluated on MNIST.

| $T = 0.9$ | $\ell_2$ radius ($r$) | 0.5 | 1.0 | 1.5 | 2.0 | 2.5 | 3.0 | 3.5 |
|---|---|---|---|---|---|---|---|---|
| | Standard | 0.0389 | 0.0951 | 0.1550 | 0.2164 | 0.2783 | 0.3404 | 0.4029 |
| | IG-NORM | 0.0394 | 0.0957 | 0.1557 | 0.2170 | 0.2790 | 0.3420 | 0.4067 |
| | IGR | 0.0390 | 0.0952 | 0.1552 | 0.2174 | 0.2818 | 0.3477 | 0.4163 |
| $T = 0.8$ | $\ell_2$ radius ($r$) | 0.5 | 1.0 | 1.5 | 2.0 | 2.5 | 3.0 | 3.5 |
| | Standard | 0.0447 | 0.1051 | 0.1691 | 0.2345 | 0.3004 | 0.3664 | 0.4329 |
| | IG-NORM | 0.0448 | 0.1052 | 0.1692 | 0.2354 | 0.3037 | 0.3733 | 0.4456 |
| | IGR | 0.0452 | 0.1057 | 0.1697 | 0.2350 | 0.3010 | 0.3680 | 0.4365 |

Table 6: Maximum allowable perturbation size for different threshold ($T = 0.8$ and $T = 0.9$) under various choices of $\ell_2$ smoothing radii $r$ evaluated on CIFAR-10.

| $T = 0.9$ | $\ell_2$ radius ($r$) | 0.5 | 1.0 | 1.5 | 2.0 | 2.5 | 3.0 | 3.5 |
|---|---|---|---|---|---|---|---|---|
| | Standard | 0.0086 | 0.0469 | 0.0885 | 0.1322 | 0.1773 | 0.2222 | 0.2683 |
| | IG-NORM | 0.0323 | 0.0705 | 0.1104 | 0.1510 | 0.1923 | 0.2337 | 0.2749 |
| | IGR | 0.0167 | 0.0545 | 0.1032 | 0.1586 | 0.2150 | 0.2588 | 0.2805 |
| $T = 0.8$ | $\ell_2$ radius ($r$) | 0.5 | 1.0 | 1.5 | 2.0 | 2.5 | 3.0 | 3.5 |
| | Standard | 0.0128 | 0.0522 | 0.0951 | 0.1402 | 0.1866 | 0.2330 | 0.2868 |
| | IG-NORM | 0.0343 | 0.0742 | 0.1157 | 0.1580 | 0.2009 | 0.2439 | 0.2867 |
| | IGR | 0.0237 | 0.0693 | 0.1258 | 0.1861 | 0.2546 | 0.3090 | 0.3559 |

Table 7: Maximum allowable perturbation size for different threshold ($T = 0.8$ and $T = 0.9$) under various choices of $\ell_2$ smoothing radii $r$ evaluated on ImageNet.

| $\ell_2$ radius ($r$) | 0.5 | 1.0 | 1.5 | 2.0 | 2.5 | 3.0 | 3.5 |
|---|---|---|---|---|---|---|---|
| $T = 0.9$ | 0.0046 | 0.0100 | 0.0152 | 0.0295 | 0.0494 | 0.0628 | 0.0768 |
| $T = 0.8$ | 0.0058 | 0.0127 | 0.0196 | 0.0369 | 0.0618 | 0.0820 | 0.1040 |

Table 8: Empirical cosine similarity between original and perturbed smoothed attributions under various choices of $\ell_2$ smoothing radius $r$, and the perturbation size computed in Table 5 ($T = 0.8$).

| $r$ | 0.5 | 1.0 | 1.5 | 2.0 | 2.5 | 3.0 | 3.5 |
|---|---|---|---|---|---|---|---|
| Standard | 0.8636 | 0.8522 | 0.8347 | 0.8127 | 0.8477 | 0.8603 | 0.8310 |
| IG-NORM | 0.8308 | 0.8181 | 0.8504 | 0.8728 | 0.8502 | 0.8193 | 0.8199 |
| IGR | 0.8231 | 0.8800 | 0.8720 | 0.8603 | 0.8362 | 0.8135 | 0.8567 |

Table 9: Minimum smoothing radius requires to achieve the threshold ($T = 0.8$ and $T = 0.9$) under various choices of $\ell_2$ perturbation size $\varepsilon$. IG-NORM and IGR are omitted since they are not scalable to ImageNet.

| | | MNIST | | CIFAR-10 | | ImageNet | |
|---|---|---|---|---|---|---|---|
| | perturbation size ($\epsilon$) | 0.5 | 1.0 | 0.5 | 1.0 | 0.5 | 1.0 |
| $T = 0.9$ | Standard | 5.1902 | 5.8752 | 5.9752 | 7.9504 | 74.6272 | 149.2544 |
| | IG-NORM | 5.1189 | 5.7699 | 5.6860 | 7.3720 | / | / |
| | IGR | 5.0265 | 5.6623 | 5.2895 | 6.5790 | / | / |
| $T = 0.8$ | Standard | 3.8927 | 4.4064 | 5.7082 | 7.4164 | 48.2095 | 96.4190 |
| | IG-NORM | 3.8392 | 4.3274 | 5.4875 | 6.9750 | / | / |
| | IGR | 3.7699 | 4.2468 | 5.0287 | 6.0573 | / | / |

allowable perturbation size is also larger. When the threshold requirement is stricter, the maximum allowable perturbation size is smaller, which suggests weaker attacks are allowed. The method is also scalable to ImageNet, which takes around 15 seconds to compute for each sample. Moreover, we also applied attribution attacks using the same radius and maximum perturbation size $\epsilon$, computed using Eqn. (8). Similar to the experiments in Section 5, we performed 20 attacks on each sample. We found that out of the total 200,000 attacked samples, the cosine similarities between clean and perturbed attributions were higher than the given threshold, suggesting that the computed bound is valid (see Table 8).

We also evaluate the third formulation that the minimum radius of smoothing required such that, within the given perturbation sizes, the cosine similarity between original and perturbed smoothed attributions is larger than the given threshold. In Table 9, the computed minimum radius of smoothing is reported. Similarly, we observe that the minimum radius of smoothing is larger when the threshold requirement is stricter, and when the attack is stronger. This is also consistent with our theory. We also notice that the radius for ImageNet is extremely large, which indicates that ImageNet is difficult to defend under such strict threshold requirements.

