# OpenReview forum: "Certified  $\ell_2$ Attribution Robustness via Uniformly Smoothed Attributions"
_ICLR.cc/2025/Conference — ICLR 2025 Conference Withdrawn Submission_

### Official Review · Reviewer_8nDD · 2024-10-24

**Soundness:** 3
**Presentation:** 3
**Contribution:** 2
**Rating:** 6
**Confidence:** 4

**Summary:**

This paper proposes using randomized smoothing with a uniform distribution to attain certified robustness for model attributions. Specifically, it proves that for all perturbations within the attack region, the cosine similarity between the uniformly smoothed attribution of a perturbed sample and that of an unperturbed sample is guaranteed to have a lower bound. The strength is that this is the first work that provides a theoretical guarantee for attribution robustness, which addresses the challenge lies in extending the randomized smoothing certification to cosine similarity.

**Strengths:**

1. Moderate Novelty: This paper achieves a notable milestone as the first work to provide a theoretical guarantee for attribution robustness. Unlike vanilla randomized smoothing, this work specifically derives robustness certification for cosine similarity, addressing a non-trivial problem.
2. Extensibility: The robustness certification for cosine similarity presented in this paper has the potential to be extended to other domains.
3. Clarity and Organization: The writing of this paper is well-organized and easy to understand.

**Weaknesses:**

1. Narrow Focus: The scope of the paper is relatively narrow, concentrating solely on extending randomized smoothing to a specific form of function—cosine similarity. Furthermore, the robustness certificate derived is limited to the uniform distribution.

2. Lack of Citation: The paper introduces the use of uniform distribution smoothing, which was first proposed in the work titled "Tight Certificates of Adversarial Robustness for Randomly Smoothed Classifiers" (NeurIPS 2019). However, this prior work is not cited. Additionally, some parts of this paper's proof bear similarities to those in the aforementioned work.

[1] Tight Certificates of Adversarial Robustness for Randomly Smoothed Classifiers. NeurIPS 2019.

**Questions:**

1. Could the robustness certification be extended to encompass more random distributions, such as the Gaussian distribution? If not, please discuss the main challenges associated with such an extension.
2. The authors suggest that cosine similarity is more likely to extend to other domains. A recommendation would be to conduct experiments in practical scenarios where cosine similarity robustness is particularly useful. Additionally, the title "Uniformly Smoothed Attributions" might be perceived as somewhat narrow in scope. Consider a title that better reflects the broader applicability of your work.

---

### Official Review · Reviewer_GjVc · 2024-10-31

**Soundness:** 3
**Presentation:** 3
**Contribution:** 3
**Rating:** 6
**Confidence:** 4

**Summary:**

This paper proposed a uniform smoothing technique to enhance the robustness of attribution methods. The authors proved a lower bound of $T$ on the cosine similarity between the uniformly smoothed attribution results of perturbed sample and the uniformly smoothed attribution results of the original sample, as stated in Eq. (6). Experiments on three datasets validated the derived theoretical bound.

**Strengths:**

1.	The paper is well-written, and the derivations are easy to follow.
2.	The paper gives a theoretical bound on the robustness of uniformly smoothed attribution technique by computing the lower bound of the cosine similarity between the uniformly smoothed perturbed sample and the uniformly smoothed original sample.

**Weaknesses:**

1.	I am not sure about the significance of attack and defense for attribution methods. In my opinion, attribution methods are targeted at specific samples and models, implying that alterations in the sample will lead to changes in the attribution results. Typically, generating adversarial examples is regarded as an attack on neural networks, as these examples possess a ground-truth label, i.e., we expect the model to classify them correctly, which it fails to do. However, for an attribution method $g$, what constitutes its ground-truth attribution map?

$\bullet$ Given a sample $x$ and a trained model $f$, the model output is $f(x)$, and the attribution result is $g(f(x))$.

$\bullet$ Given a seemingly similar adversarial sample $x_{\text{adv}}$, the model output is $f(x_{\text{adv}})$, and the attribution result is $g(f(x_{\text{adv}}))$.

In above scenarios, it is acceptable for the attribution results of samples $x$ and $x_{\text{adv}}$ to differ significantly, as the model output has changed considerably.

Similarly, for a sample $x_{\text{noise}}$ with general additive noise (e.g., Gaussian noise or uniform noise), where the model output is $f(x_{\text{noise}})$ and the attribution result is $g(f(x_{\text{noise}}))$, if the logit value changes to some extent without altering the classification result, then variations in the attribution result should also be acceptable. Why is this considered an attack, and why is there a need to defend against such an attack?

To improve it, the authors should elaborate on specific use cases where the robustness of attribution is important, even in the absence of an explicit ground-truth attribution map. Besides, in this context, the authors should discuss how to define a "valid" or "meaningful" attribution, and why the stability of attribution might be desirable in certain applications.

2.	Why is the expectation of the attribution maps for perturbed samples in Eq. (2) considered the ideal attribution result? Consider the extreme case: if we average the attribution results $g(f(x))$ of a correctly classified original sample $x$ and the attribution results $g(f(x_{\text{adv}}))$ of a misclassified adversarial sample $x_{\text{adv}}$, what is the physical meaning of this average result? It appears that this does not accurately represent the attribution result for sample $x$, nor for the adversarial sample $x_{\text{adv}}$. To explain this, the authors may provide empirical or theoretical justification for why the smoothed attribution is a meaningful or useful metric.

3.	How is the Lipschitz constant in Lemma 1 determined? Specifically, how can we theoretically and practically identify the maximum of the gradient's Lp norm, $||\bigtriangledown f(x)||_p$, for all $x\in S$? For instance, in the context of the integrated gradients method, what would be the maximum value of $||\bigtriangledown f(x)||_p$? In this case, what is $M$ in Theorem 1? The authors could provide specific examples of how these values are computed or estimated for typical network architectures or attribution methods. Additionally, the authors may discuss any assumptions or approximations made in determining these values in practice.

4.	Does Theorem 1 apply specifically to gradient-based attribution methods, or can it also be extended to perturbation-based attribution methods? What are the respective upper bounds $M$ for the attribution results in these two categories? Is $M$ a theoretical value (e.g., the Lipschitz constant of a general DNN), or is it an empirically measurable value (e.g., the maximum gradient of a specific sample)? If there exist any limitations to the other types of attribution methods, the authors can describe them. Besides, the authors should provide a more detailed discussion of how M is determined in practice for different types of attribution methods.

**Minor:**

In Line 404-405, the authors choose the number of samples $N$ to be 100, 000 to compute the lower bound. How should $N$ be selected for different tasks? It would be beneficial to include an empirical analysis on a dataset, such as MNIST, showing how the theoretical lower bound $T$ in Eq. (6) varies with different values of $N$. The authors could provide guidelines or empirical results on how to choose an appropriate value of N for different datasets or tasks, and to show how the bound varies with N.

**Questions:**

1.	According to Weakness 1, is it necessary to impose constraints on the model output $f(x)$ in Eq. (1), such as requiring that $f(x)$ and $f(x + \delta)$ are similar or yield the same classification result?

2.	In Line 751, do R3 and R1 represent the centers of the spheres for $g(\tilde{x})$ and $g(x)$, respectively?

---

### Official Review · Reviewer_88rx · 2024-10-31

**Soundness:** 2
**Presentation:** 1
**Contribution:** 3
**Rating:** 1
**Confidence:** 4

**Summary:**

This work presents a method for providing certificates on attribution robustness under $\ell_2$ attacks. Specifically, the paper presents a method for computing the smoothed attribution robustness of a neural network. This allows them to handle networks not certifiable by previous methods which require twice-differentiability. The paper also reformulates the certificates allowing it to certify networks on minimum attack radius or maximum attack radius given uniform sampling density radius. Finally, the paper provides a wide range of experiments on different network architectures for MNIST, CIFAR10, and ImageNet showing the effectiveness of their method.

**Strengths:**

Providing certificates for smoothed attributions is a novel idea to the best of my knowledge, and I think it could be valuable for larger networks or on networks current methods cannot handle. I find the paper mostly flows aside from the comments I made below, and the proofs seem to be correct to me aside from some small typos. The alternative formulation of the certificate seems to me like it would be useful in many instances. The experiments section/appendix give a fairly large number of diverse experiments showing the method's ability to be used for different networks, datasets, etc.

**Weaknesses:**

> To the **AC, Other Reviewers, and Authors** - I have major concerns with the writing quality of this paper. In the introduction/abstract alone I think there are at least 48 issues with clarity/grammar (see my questions below - Clarity & Minor Comments). Usually, I leave these comments for the authors and they do not affect my rating; however, while this paper is understandable the amount of these issues makes it very hard to read.

In the questions section below I detailed 5 key concerns I have with the paper. I summarize them here, but refer to the below section for more detailed questions.

1. The certificates in this paper seem to be neither deterministic nor probabilistic certificates, they depend on sampling to estimate expectation but there is not guarantee given on the probability given the number of samples taken that I see.
2. It is unclear why a uniform distribution is used rather than gaussian like in some similar works.
3. The paper claims it works for all NN architectures, but fails to mention it assumes Lipschitz continuity.
4. To the best of my understanding, the certificates depend on computing the Lipschitz constant of the network; however, it is never mentioned how this is done.
5. There is almost no discussion on runtime, and the one number given is confusing to me.

Finally, I found myself constantly referring back and forth to the Appendix to get certain definitions of variables to really understand the points being made. There also seems to be a fair number of variables that the reader must assume the definition of or must dig to find the definition of.

**Questions:**

1. There is a brief mention (paragraph starting at 398) that the certificates provided in this paper are actually 'probabilistic' certificates as $h(x)$ cannot actually be exactly computed. This needs to be made much more explicit. Compounding this is the fact that paragraph just mentions the number of samples used for the experiments but does not ever mention the 'actual probability' that these certificates hold at. In Appendix C.1 the paper empirically shows that when $N = 100,000$ that the certificates are *close* for the networks the datasets considered when $\alpha = 0.01$ (i.e. the certificate *almost* holds with probability $99\%$ when using 100,000 samples). In other words, the computed values in this paper are neither a true certificate nor a probabilistic certificate (line 403: "almost surely"). This leads into my next question as it should be possible to obtain true probabilistic certificates.
2. There is an unsubstantiated claim in line 195 that because this paper aims to certify under $\ell_2$ attacks a uniform distribution should be chosen. However, in the original randomized smoothing paper (Cohen et al. 2019) $\ell_2$ attacks are being certified but the distribution chosen is Gaussian. See Theorem 2 in Cohen et al. 2019: "Theorem 2 shows that Gaussian smoothing naturally induces $\ell_2$ robustness: if we make no assumptions on the base classifier beyond the class probabilities (2), then the set of perturbations to which a Gaussian-smoothed classifier is provably robust is *exactly* an $\ell_2$ ball." What happens if you use a gaussian distribution instead? What is the benefit of a uniform distribution?
3. "The proposed result does not assume anything on the classifiers" (line 308) is contradicted by "is a function of the input gradient, $\nabla f(\mathbf{x})$, which is bounded for Lipschitz continuous networks" (line 261). While *most* networks are Lipschitz continuous, not all networks are. For example, "The Lipschitz Constant of Self-Attention" Kim et al. Theorem 3.1 gives Dot Product Multihead Attention is not Lipchitz continuous.
4. As a followup to weakness 3, the computations in this paper depend on $M$ which is defined as $||g(x)||_2 \leq M$ (line 756, also for an important variable why is it only formally defined in the appendix). $g$, although never formally defined, is a function of $\nabla f(\mathbf{x})$. $\nabla f(\mathbf{x})$ is bounded by the Lipschitz constant of the network. There are many potential ways to compute the Lipschitz constant of the network, is the paper using an approximate method? Which one? It is important to know as it affects both the precision of the certificate and the runtime.
5. On the note of runtime, I could be wrong but in the entire paper and appendix I only found a single mention of runtime at line 475 which claims that certification for ImageNet on ResNet-50 takes around 15 seconds per sample. According to line 405, the number of samples is 100,000. Even being generous this seems slightly hard to believe to me. This is not scientific, but if we look at some online resources [1] with half-precision shows 4,000 inferences per second on a 3080 GPU, a comment in [2] indicates that the 3090 is roughly 30% faster than the 3080 on ResNet-50 inferences. This means in 15 seconds you can do roughly 80k inferences. However, all the attribution methods (including the ones used in this paper) require backpropping all the way to the input layer, an additional cost on top of inference time. This also does not include Lipschitz constant computation time [3] may not be the most up to date but it suggests that even approximations should take a couple of seconds if not tens-hundreds of seconds. I would like to see a more comprehensive analysis of runtime for the experiments because the numbers are slightly confusing me.

[1] https://medium.com/@niksa.jakovljevic/optimizing-resnet-50-8x-inference-throughput-with-just-a-few-commands-41fa148a25df

[2] https://www.reddit.com/r/deeplearning/comments/zf3qn3/rtx_3080_vs_rtx_3090/#:~:text=In%20DL%20benchmarks%2C%20even%20if,because%20of%20the%20memory%20limitations.

[3] https://proceedings.neurips.cc/paper_files/paper/2019/file/95e1533eb1b20a97777749fb94fdb944-Paper.pdf

**Suggestions**

1. Can you define attribution function $g$ or at least verbally describe it in the paper?
2. Can you boost the brightness of Figure 1 (b, c, d)? These will not look good especially when printed.
3. $cos$ is a different function from cosine similarity, please define it.
4. What do the three numbers in line 201 represent, do we want higher values? What do they mean? How do they relate to effectiveness of attribution?
5. Why is it intractable to optimize cosine similarity (line 228)? Can you cite something or prove it?
6. $a\mathbf{v} = R_1 - R_3$ line 752
7. line 242 this arbitrary translation can be made regardless of what you have written above, but what you have written above is also not enough to show that the magnitude of $a$ is constrained by "a constant related to the intersecting...". Explicitly say here what $a$ is restrained by you can leave the proof in the appendix.
8. line 248, 782 what about $h(\mathbf{x})$ and $h(\mathbf\{\tilde{x}})$ form a spherical cone? Neither the intersection, union, nor symmetric difference form a spherical cone (https://en.wikipedia.org/wiki/Spherical_sector), or if you say they do, can you prove it? What does the spherical cone have to do with directional decomposition?
9. line 405, I don't think $N^*$ is defined.
10. line 706, $q$ -> $p$

**Related Work**

1. As you are only required to address articles published before July 2024, this does not count towards my grade; however, I am curious how your method compares to "Advancing Certified Robustness of Explanation via Gradient Quantization" CIKM '24 which uses ideas from randomized smoothing to provide explanation robustness.

> Below, I give comments on clarity and writing. I only have time to do this for the abstract and introduction given the amount of issues, please go over the entire paper making these corrections.

**Clarity**

1. (line 17) "from a certain space" is too ambiguous
2. (line 18) "the attack region"? What attack region, you haven't defined that.
3. (line 19) "guaranteed to be lower bounded." Lower bounded by what? I can prove that it is lower bounded by $-1$ but that is not interesting, what do you lower bound it by? either what method, or what metric, etc?
4. (line 20) "of the certification" what certification? "the original one" what original one? maybe you are trying to say something like "**Additionally, we certify the minimum smoothing radius (or maximum perturbation size) such that an attribution remains unchanged.**"
5. (line 23) "the proposed method" what proposed method? There is a derivation of a formulation, and a proof what is the method?
6. (line 23) "on three datasets" be clear here, what datasets?
7. (line 33) What applications?
8. (line 40) "people naturally requires the confirmation of the safety critical decisions" does not make sense, I assume you are trying to say that people would want to know how safety critical decisions are made but it is unclear
9. (line 41) The risk is because the users lack expertise in ML and technical details of attributions, or the risk is because the model is not robust. Even if a user was an expert in ML/attributions an adversarial attack changing the attribution would likely be hard to notice.
10. (line 44) In contrast to the above comment, shouldn't it be "users" who lose trust?
11. (line 45) "Thus, a trustworthy..." does not follow from the previous statements, I think you want to say that a trustworthy model must produce both robust predictions but also robust explanations. Also, there are other methods of explanation besides attribution so a trustworthy model could have explainability produced by a different method.
12. (line 53) "Unlike adversarial defense, which has been ..." This sentence has multiple issues, "that using both empirical..." comes after adversarial attacks, except what I assume you mean is that there are both empirical and certified adversarial defense methods. Instead I think you want to say something like "While there has been extensive work in both empirical (cite) and certified (cite) defenses against adversarial attacks targeting prediction accuracy, there are almost no works on defending against attacks targeting model attributions.
13. (line 60) What is categorical ranking measurements? Why are they not easy to extend to other domains?
14. (line 61) Do they attempt to derive, or they do derive? I think you want to say something like "Wang & Kong (2023) derives a practical upper bound of the cosine similarity of the worst-case attribution under attack; however, their method has strict assumptions <say on what> and is computationally intensive"
15. (line 63) "the smoothed version of attributions" does not follow because you have not described a smooth version of model attribution. "we focus on smoothed model attributions and provide a theoretical guarantee that these attributions are robust to any perturbation within an $\ell_2$ ball.
16. (line 75) "we find an effective upper bound on Equation (1), or equivalently, a lower bound on the worst-case attribution similarity." I assume you do find a bound, not attempt to find a bound.
17. (line 76) "We propose.." This sentence is way to long and does not read well. "We propose uniformly smoothed attribution - a smoothed version of model attribution. We analytically provide a lower bound on the worst case cosine similarity for uniformly smoothed attribution."
18. (line 82) The theoretical guarantee demonstrates? "We provide a theoretical guarantee on the robustness of uniformly smoothed attribution under perturbations belonging to a given input specification."
19. (line 83) "The robustness is measured by..." This is not new, Equation (1) from Wang & Kong is the same, you don't need this sentence.
20. (line 85) Just larger sized images? or also larger and more diverse network architectures?
21. (line 86) Wang & Kong provides theoretical guarantees (see theorems 1 and 2), so this is not the first work that provides theoretical guarantees for attribution robustness. Instead you maybe want to say that it is the first scalable or the first work that provides guarantees for general NN architectures.
22. (line 88) Your alternate certificates are not equivalent, the original ones have a bound on the perturbation size, this certificate gives the maximum perturbation radius for which the attribution difference remains under a certain threshold. I understand that they are derived in a similar way, but they give different guarantees.
23. (line 93) "Well-bounded integrated gradients" <- you never define this, cite the paper or something? what does it mean for it to be well-bounded?
24. (line 99) "this paper and" -> "the paper"

**Minor Comments**

1. (line 12) "the attributions" I think should be "**model** attributions", similarly "... manipulate **model** attributions **without changing model predictions**"
2. (line 15) "an effective **verification** method" or "an effective **certification** method" since you are using these to understand robustness not defend against it
3. (line 17) "use **a** uniform smoothing"
4. (line 17) "It is proved" i.e. someone else did it, if its this case, say who? or "**We proved**" i.e. you do it in this paper
5. (line 19) "the cosine similarity between **the** uniformly smoothed attribution of **a** perturbed sample and **an** unperturbed sample is..."
6. (line 24) "the attributions"
7. (line 25) networks -> network, schemes -> scheme, datasets -> dataset
8. (line 29) developments -> development
9. (line 30) have -> has, discussions of the -> discussion on
10. (line 32) (Brown et al, 2020) links to "Language models are few-shot learners." what does this paper have to do with explainability of LLMs?
11. (line 38) "the explainability" or what? or what explainability?
12. (line 40) requires -> require
13. (line 46) "the attributions" -> "model attributions", have also been shown recently -> have recently been shown to also
14. (line 48) can be -> are
15. (line 50) "This misleadingly..." this sentence is basically already said above
16. (line 63) "Currently, there is no scalable method for providing generalized certificates of attribution robustness"
17. (line 64) we put our focus -> we focus
18. (line 67) "Based on the previously..." -> "Wang & Kong (2023) define attribution robustness as the worst cases attribution difference $D(\cdot, \cdot)$ resulting from a perturbation within a certain attack radius. Formally, given a network $f$, ..., Wang & Kong (2023) define attribution robustness as..."
19. (line 70) this formula does not depend on $f$, you have not defined an attribution function $g$, I think you want to somehow specify that $g$ depends on $f$ for example, $g_f$
20. (line 73) assumptions -> assumption, networks -> network, ", and, as a result, is unable to" -> "so it does not"
21. (line 74) "give a complete certificate" or "provide a completely certify model attribution for general neural networks"
22. (line 80) contribution -> contributions
23. (line 92) the attribution -> model attributions
24. (line 93) evaluated -> evaluate

---

### Official Review · Reviewer_ZeRi · 2024-11-02

**Soundness:** 2
**Presentation:** 2
**Contribution:** 2
**Rating:** 3
**Confidence:** 3

**Summary:**

Model attributions are important for security-sensitive applications. However, attribution attacks can mislead users and undermine trust in AI systems. The paper critiques existing defenses against attribution attacks, noting that most focus on adversarial training rather than directly safeguarding attributions. The authors propose a novel approach using uniformly smoothed attributions to enhance the robustness of attribution models. Their method offers a theoretical guarantee of robustness against perturbations within a defined attack budget.

**Strengths:**

* A theoretical framework demonstrating that uniformly smoothed attributions are robust to perturbations.

* Alternative formulations for certification, allowing practical implementation to determine the maximum perturbation size or the minimum smoothing radius.

**Weaknesses:**

This work represents a significant advancement in ensuring the reliability of attributions in AI systems, providing a foundational approach for further research in attribution robustness. However, I still see some issues may help to improve the quality of the work.

* There is a significant body of research focused on certifying the robustness of DNNs. How does this work differ technically from existing robustness certification approaches for DNNs? Additionally, it is unclear which specific targeted attribution models this work aims to protect. Are they saliency maps and integrated gradients? These attribution models are inherently limited in their ability to achieve high accuracy on complex tasks. The authors should provide justification that the attribution models they choose to protect play a critical role in security-sensitive applications and are widely adopted in practical scenarios. What specific real-world applications would benefit from robust attribution models? Can the authors provide examples of security-sensitive scenarios where protecting saliency maps or integrated gradients is crucial?

* The necessary preliminaries have not been introduced. For instance, the specific smooth process under a given maximum allowable perturbation $\epsilon$, is lacking. Additionally, what is the difference between the maximum allowable perturbation $\epsilon$ and $l_2$-norm ball with a radius $r$? Without these preliminaries, it is challenging to follow the authors' theoretical framework. I would like to suggest that the authors add a "preliminaries" section defining key terms and concepts, including a clear explanation of the smoothing process, the relationship between $\epsilon$ and $r$, and how these parameters are used in the theoretical framework.

* In Section 4.1, the induction process from Equation 4 to Equation 5 is not clearly articulated. Additionally, the conclusion regarding the magnitude of $a$ is also ambiguous. Specifically, it is unclear why it is bounded by the magnitude of the gradient weighted by the ratio of volume change during the translation process. The authors should provide step-by-step derivations for the transition from Equation 4 to Equation 5, and clearly explain the reasoning behind the bound on the magnitude of  $a$. In addition, in this work, the existence of the upper bound of a given (gradient-based) attributeion model depends on the Lipschitz continuous networks assumption. Please justify the Lipschitz continuity assumption and discuss how their method might be adapted for non-Lipschitz networks.

* In Section 4.3, the relationship between the maximum allowable perturbations $\epsilon$ and the threshold distance between the input and its smoothed version (as shown in Eq. 8) is evident. However, the role of the additional variable $R$ in Eq. 9 is unclear. What does the variable $R$ represent in Equation 9?

* There are other $l_p$-norms that can measure the distance between the original input and its perturbed version. Why do the authors only consider the $l_2$ attribution attack? The authors should discuss the applicability of the proposed theorem to other $l_p, p=0,\infty$ attribution attacks. Additionally, why is the density $\mu$ in this work chosen to be a uniform distribution rather than another distribution, such as Gaussian or Poisson? The authors should explain their rationale for focusing on $l_2$-norm and uniform distributions. They should also discuss whether their approach could be extended to other norms and distributions, and if so, what modifications would be required.


* Minors:
  * In section 2.3, the concepts of randomized smoothing and uniform smoothing, may easily be confused.
  * Tables 1 and 2 are located far from their corresponding sections, making it inconvenient to observe the data.

**Questions:**

* What is the difference between the maximum allowable perturbation $\epsilon$ and $l_2$-norm ball with a radius $r$?

* How do we use the proposed method if the gradients do not satisfy the Lipschitz condition?

* What does the variable $R$ represent in Equation 9?

* Why do the authors only consider the $l_2$ attribution attack?

* Why do the authors choose uniform distribution?

---

### Note · Authors · 2024-11-17

I have read and agree with the venue's withdrawal policy on behalf of myself and my co-authors.